# Relationship between Clinical Parameters and Chromosomal Microarray Data in Infants with Developmental Delay

**DOI:** 10.3390/healthcare8030305

**Published:** 2020-08-27

**Authors:** Zeeihn Lee, Byung Joo Lee, Sungwon Park, Donghwi Park

**Affiliations:** 1Department of Rehabilitation Medicine, Daegu Fatima Hospital, Daegu 41199, Korea; zilee@hanmail.net (Z.L.); bjl84@naver.com (B.J.L.); 2Department of Physical Medicine and Rehabilitation, Ulsan University Hospital, University of Ulsan College of Medicine, Ulsan 44033, Korea

**Keywords:** chromosomal microarray, dysmorphism, developmental delay, brain and cardiac anomalies, genetic counseling

## Abstract

Chromosomal microarray (CMA) is considered a first-tier test for genetic analysis as it can be used to examine gene copy number variations (CNVs) throughout the entire genome, with enhanced sensitivity for detecting submicroscopic deletions and duplications. However, its cost can represent a heavy burden. Moreover, the diagnostic yield of CMA in infants with developmental delay (DD) was reported to be less than 10%. Therefore, we aimed to investigate the relationship between CMA results and clinical features and risk factors of DD. The study included 59 infants with DD who were recruited between August 2019 and February 2020 during a visit to the outpatient clinic of a rehabilitation department. We reviewed the clinical records of the infants regarding gender, age, body weight at birth, delivery method, brain imaging data, perinatal history, and parent-related clinical parameters, such as mother and father age at birth. The infants were categorized according to CMA results, and differences in clinical parameters were evaluated. Except for brain anomalies, there was no statistically significant differences between infants who had pathogenic and variants of unknown significance (VOUS)-likely pathogenic CNVs groups compared with those within the VOUS-likely no sub-classification, VOUS-likely benign, benign, and normal CNVs groups. The incidence of brain anomalies was significantly higher within infants with pathogenic and VOUS-likely pathogenic CNVs groups (*p* < 0.05). Our study suggests that infants with DD who present dysmorphism or brain anomaly may benefit from early CMA analysis, for adequate diagnosis and timely treatment. Further studies are warranted to confirm the relationship between DD clinical parameters and CMA results.

## 1. Introduction

A developmental delay (DD) is characterized by noticeable shortage of developmental milestones achieved by a child throughout development [1]. Although there is no exact definition, the DD concept is generally used to describe delays in major criteria of the developmental process, including cognitive, physical, communication, emotional, and social impairments [1]. DD incidence rate in the general population is close to 3% [2,3]. Early screening of DD may represent an important strategy to support early therapeutic intervention and enhance the prognosis of those affected [4,5]. However, only 30% of DD affected children are diagnosed at pre-school age [6]. In many cases, late recognition can be explained by the absence of definite clinical symptoms of DD and its comorbidities, making it difficult for caregivers and parents to perceive this condition and seek clinical help [7]. Recent findings on genetic defects in infants with unexplained DD have provided important clues on the diagnostic value of screening tests for early diagnosis of DD [8].

Several cytogenetic and molecular biology techniques are used to identify the underlying genetic cause of DD, from conventional approaches (such as molecular karyotyping) to state-of-the-art chromosomal microarray (CMA) and next-generation sequencing (NGS) [9]. Recently, CMA has been widely used as a first-tier of genetic analysis in place of G-banded karyotyping as it can examine gene copy number variations (CNVs) throughout the genome with improved sensitivity for submicroscopic deletions and duplications [9,10]. However, this diagnostic improvement is associated with a significant economic burden to the caregivers [9]. Nevertheless, Farooqi et al. showed that the diagnostic yield of CMAs, with identification of pathogenic CNVs or variants of unknown significance (VOUS)-likely pathogenic, was less than 10% in infants with DD [11]. This finding highlights the clear need of a better understanding of DD clinical features and its association with CMA data to reduce unnecessary CMA testing. In this study, we investigated the correlation between various clinical parameters, including risk factors of DD, and CMA results collected from infants with DD.

## 2. Methods

### 2.1. Infants with DD

The study included 59 infants who visited the outpatient department of rehabilitation medicine at our hospital from August 2019 to February 2020 for suspected DD. Most of the infants showed delayed gross motor function or impaired language performance compared to age-matched children with no developmental problems. According to the after Korean Developmental Screening Test (K-DST) manual, the children can be categorized into four groups based on their individual K-DST result: further evaluation (<−2 standard deviation [−2SD]), follow-up test (−2SD to −1SD), peer-level (−1SD to 1SD), and high level (>1SD) [12,13]. We recommended CMA testing for children classified within the ‘further evaluation’ or ‘follow-up test’ groups. The K-DST includes 6 sections that assess developmental areas of gross motor, fine motor, cognition, communication, social interaction, and self-control. Additional questions are designed to take into account clinically important diseases, such as cerebral palsy, language delay, and autism [12,13]. Therefore, all infants with DD who included in this study were classified within the ‘further evaluation’ or ‘follow-up test’ groups. We also reviewed the clinical records of the infants regarding their sex, age, body weight at birth, gestational age, delivery method, perinatal history, as well as parent-related clinical parameters, including mother and father age at the time of the child birth. This study was approved by the Institutional Review Board of our hospital (DFE200RIO074.). A detailed description of the CMA test and an agreement to perform the test was obtained from the parents of all infants. The parents also agreed to release the results of the CMA test for research purposes.

### 2.2. CMA Protocol

Peripheral venous blood was collected into an ethylenediaminetetraacetic acid (EDTA) collection tube, and DNA was extracted from peripheral blood leukocytes. CytoScanDx assay (Affymetrix, Santa Clara, CA, USA) was used for CNVs analysis and it was performed according to the instructions of the manufacturer [10]. The assay comprises more than 750,436 CNV markers, including 200,436 genotype-able single nucleotide polymorphisms probes and more than 550,000 non-polymorphism probes. The overall average marker space was 4127 base pairs [10]. All data were visualized and analyzed through the GeneChip System 3000Dx (Affymetrix) platform using the human genome assembly GRCh37 (hg19) as reference [10]. All deletions and duplications with more than 400 kb were reported, and those with 400 kb or less, well-known microdeletion, duplication syndrome sites, or affecting clinically relevant gene sites were also reported [14]. The CMA results were reported according to the following classifications: pathogenic CNVs, variant of unknown significance (VOUS)-likely pathogenic CNVs, VOUS with no sub-classification, VOUS-likely benign, benign, or normal CNVs [10].

### 2.3. Statistical Analysis

The Kolmogorov–Smirnov test was used to determine whether the data conformed to a normal distribution. Pearson Chi-square test, independent *t*-test, and Mann–Whitney U-test were used to evaluate the differences between pathogenic, VOUS-likely pathogenic CNVs, and other groups, and between two groups (pathogenic, VOUS with no sub-classification, VOUS-likely benign versus benign and normal CNVs group). Categorical variables included gender, brain anomalies, cardiac anomalies, dysmorphism, autism spectrum disorder, cerebral palsy, delivery method, and mode of conception. Continuous variables included age, father and mother age, body weight at birth, and gestational age. Statistical analyses were performed using SPSS 25.0 for Windows (IBM, Armonk, NY, USA) and R software for Windows (version 2.15.2; R Foundation for Statistical Computing, Vienna, Austria). A *p*-value < 0.05 was considered to be statistically significant in all tests.

## 3. Results

### 3.1. Demographic and Genetic Characterization of the Patients

CMA data were collected from 59 infants, of whom 41 were males and 18 were females. All 59 infants with DD who included in this study were classified within the ‘further evaluation’ or ‘follow-up test’ groups according to K-DST (‘further evaluation’: 23 infants; ‘follow-up test’: 36 infants). The average age of the infants was 38.7 ± 18.50 months, and the average age of their fathers and mothers at the time of birth was 35.7 ± 5.23 and 32.2 ± 4.04 years, respectively. The average birthweight of the infants was 2921.0 ± 716.48 g, their gestational age was 37.4 ± 2.72 weeks, and 61% of children were born by cesarean section (Table 1).

We found 41 CNVs in 35 infants, including 4 (9.8%) pathogenic CNVs, 2 (4.9%) VOUS-likely pathogenic CNVs, 16 (39.0%) VOUS with no sub-classification, 16 (39.0%) VOUS-likely benign CNVs, and 3 (7.3%) benign CNVs (Figure 1). CMA results and affected genes, clinical features, and other risk factors are summarized in Table 2. Typical genetic problems that were identified among pathogenic CNVs, such as Klinefelter’s syndrome, mosaicism of Turner’s syndrome, or genetic problems associated with incest history, are also reported. The remaining 24 infants had normal CNV findings.

### 3.2. CMA Results Comparisons

No significant difference was observed between pathogenic and VOUS-likely pathogenic CNV groups compared with VOUS with no sub-classification, VOUS-likely benign, benign, and normal CNVs groups, except for brain anomalies (Figure 2). Significantly more infants classified within the pathogenic and VOUS-likely pathogenic CNV groups had brain anomalies (*p* < 0.05). Moreover, the rate of infants with cardiac and/or brain anomalies, and dysmorphism was higher in the pathogenic and VOUS-likely pathogenic CNV groups than in the VOUS like no sub-classification, VOUS-likely benign, benign, and normal CNVs groups. However, this difference did not reach statistical significance (*p* = 0.053) (Figure 2). Additional analyses showed that gender, age, body weight at birth, gestational age, delivery method, mode of conception, result of brain magnetic resonance imaging, and perinatal history of the infants, as well as parent-related clinical parameters, were that there were no significant difference observed between pathogenic and VOUS-likely pathogenic CNV groups compared with VOUS with no sub-classification, VOUS-likely benign, benign, and normal CNVs groups, except for brain anomalies (Figure 2).

Between pathogenic, VOUS-likely pathogenic, and VOUS with no sub-classification versus VOUS-likely benign, benign, and normal CNVs groups, additional analyses showed that gender, age, body weight at birth, gestational age, delivery method, mode of conception, result of brain magnetic resonance imaging, and perinatal history of the infants, as well as parent-related clinical parameters, showed no significant differences (Figure 3).

## 4. Discussion

We found four pathogenic CNVs and two VOUS-likely pathogenic CNVs in six infants, which indicated that 10.2% of the infants had genetic alterations associated with DD and comorbid conditions. This finding was in concordance with the findings of a previous CMA study conducted in infants with DD [11].

We identified a variety of CMA results; however, most of these genetic alterations were of unknown phenotype (39% of VOUS with no sub-classification). Nevertheless, we expected to find more abnormal CNVs within our study population. Therefore, we also evaluated potential correlations between demographic and clinical parameters of the infants and their parents, including risk factors known to cause DD, and the genetic findings. This analysis failed to find significant differences between pathogenic and VOUS-likely pathogenic CNV groups versus VOUS-likely no sub-classification, VOUS-likely benign, benign, and normal CNVs group, with exception of brain anomalies. Indeed, our data showed a tendency of slightly higher rate of infants with brain and cardiac anomalies and dysmorphism within the pathogenic and VOUS-likely pathogenic CNVs groups. Nevertheless, only one and two infants included in the study had brain anomalies and dysmorphism, respectively, which may contribute to the low statistical power of the analysis. Kim et al. reported that dysmorphism was significantly higher in infants with pathogenic CNVs as compared with those with normal CMA results [10]. Moreover, Gilissen et al. also reported that infants with intellectual disability and multiple congenital anomalies have higher burden of CNVs than those with intellectual disability alone [15]. Altogether, these findings support the important role of CMA for assessing DD infants with anomalies and dysmorphism.

Interestingly, the average age of infants with pathogenic or VOUS likely-pathogenic CNVs was 22.1 months, which was much lower than the overall average age of the study population (Figure 2). This result suggests that children with pathogenic or VOUS likely-pathogenic CNVs may have a more prominent developmental delay, leading them to visit the hospital earlier and to receive an earlier diagnosis, which is definitely advantageous for planning their rehabilitation program. In such cases, as well as in cases of identified VOUS no sub-classification, VOUS likely-benign, and even benign CNVs, an early diagnosis allows clinicians to discuss with the parents about existing genetic risks and provide them genetic counseling before having another child.

Despite of these findings, our study has some limitations. First, we investigated CMA results within a small number of infants with DD. However, we expanded our analysis to include additional demographic and clinical parameters, such as body weight at birth, delivery method, and parent-related clinical parameters, which was not performed in the previous study [10]. Contrary to our expectation, the age of the parents seemed to not have a significant impact on the incidence of clinically significant CNVs. Nevertheless, more CMA studies are warranted with larger cohorts of infants with DD to further confirm this finding. Secondly, although CMA offers the sensitivity of high-resolution genome-wide detection of clinically significant CNVs, there is an additional challenge for interpreting VOUS, which is the preferred terminology based on a recent study of variant terminology [9]. Moreover, there are definite limitations of CMA for genetic analysis. For example, CMA cannot detect balanced chromosome rearrangements, such as inversions or translocations, which do not result in deletion or duplication of genetic material, or cases of low-level tissue mosaicism, although balanced rearrangements rarely are associated with disease unless there is disruption of a critical gene. Additionally, CMA may not identify low levels of tissue mosaicism in the fetus [9,16]. Additionally, uniparental disomy may also not be detected when the region for which heterozygosity was lost is small or in cases of heterodisomy. Furthermore, 13p, 14p, 15p, 21p, 22p, Yq11.23, Yq12, and pericentric heterochromatin region of all chromosomes are regions that are undetectable by CMA [9,16]. Despite these technical limitations, CMA is still used as a primary method for detecting gene deletions and duplications throughout the entire genome. Therefore, our study may provide helpful information for selecting infants with DD whose families may benefit of CMA testing. More studies addressing data collected from multiple genetic analysis platforms, such as CMA and next-generation sequencing, in infants with DD maybe helpful to further confirm the relationship between genetic and clinical features of DD.

## 5. Conclusions

In conclusion, our research suggests that infants with DD who present dysmorphism or brain anomaly may benefit from early CMA analysis for adequate diagnosis and timely treatment. However, further studies are needed with focus on CMA data in infants with DD.

## Figures and Tables

**Figure 1 healthcare-08-00305-f001:**
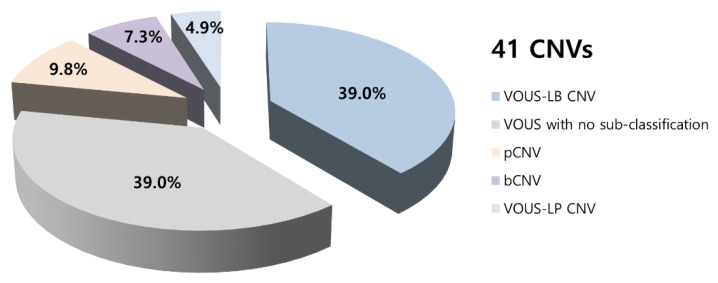
Summary of chromosomal microarray results. CNV—copy number variations, bCNV—benign copy number variations, pCNV—pathogenic copy number variations, LB—likely-benign, LP—likely-pathogenic, VOUS—Variant of unknown significance.

**Figure 2 healthcare-08-00305-f002:**
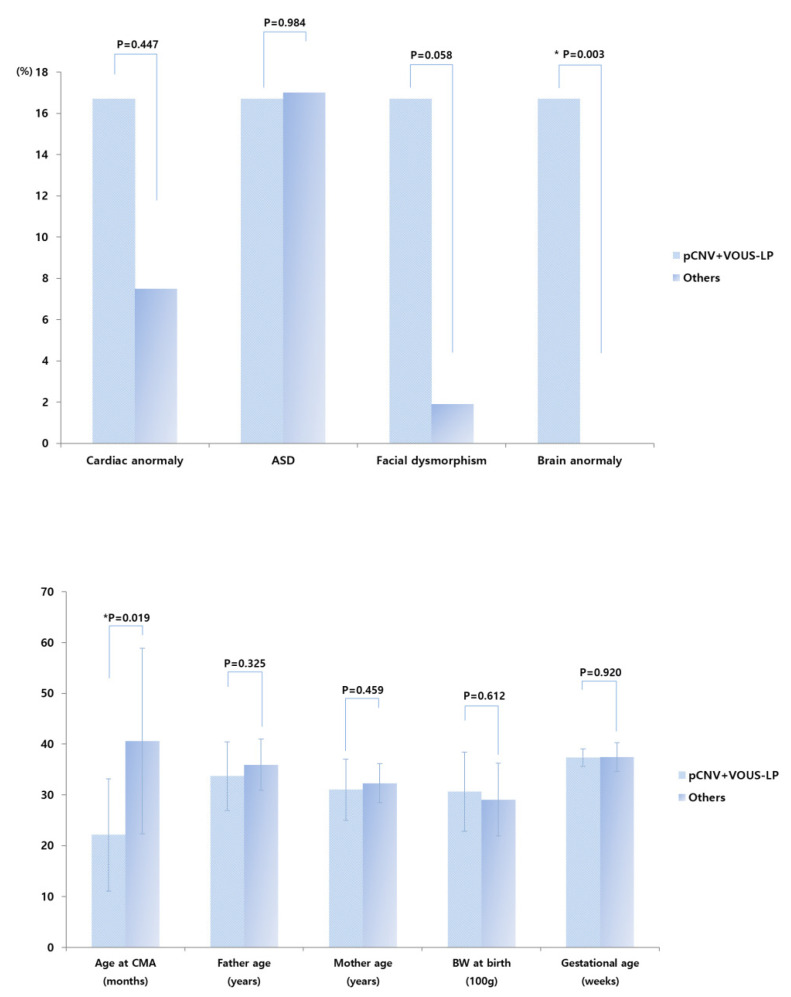
Prevalence of cardiac anomaly, autism spectrum disorder, facial dysmorphism, and brain anomaly in pathogenic and VOUS-likely pathogenic CNV groups versus VOUS with no sub-classification, VOUS-likely benign, benign, and normal CNVs groups. CMA—chromosomal microarray analysis, BW—body weight, ASD—autism spectrum disorder, pCNV—pathogenic copy number variations, LP—likely-pathogenic, VOUS—Variant of unknown significance. * *p* < 0.05.

**Figure 3 healthcare-08-00305-f003:**
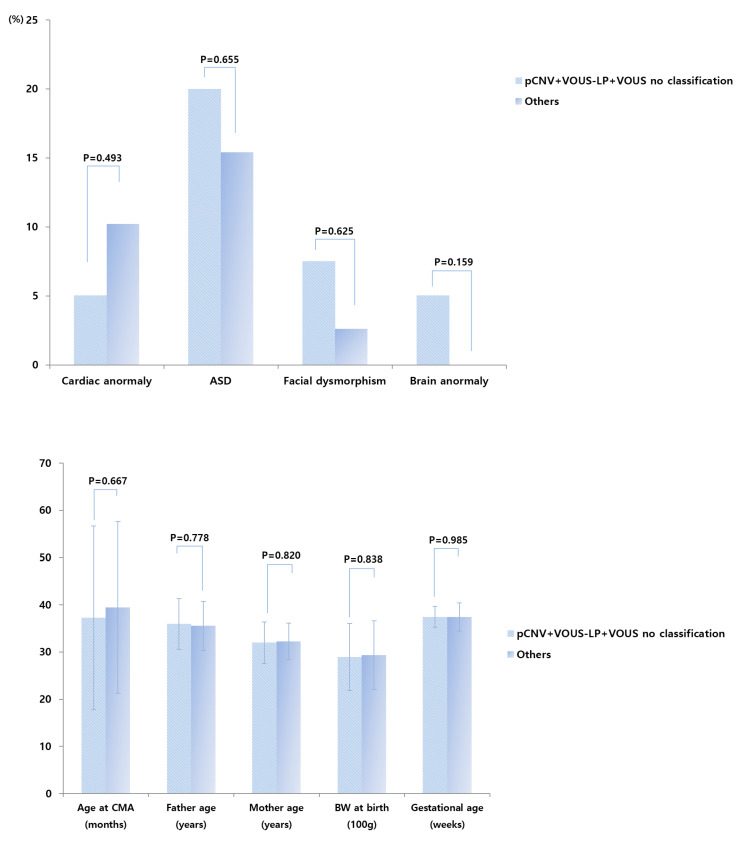
Prevalence of cardiac anomaly, autism spectrum disorder, facial dysmorphism, and brain anomaly in pathogenic, VOUS-likely pathogenic CNV groups, and VOUS with no sub-classification versus VOUS-likely benign, benign, and normal CNVs groups. CMA—chromosomal microarray analysis, BW—body weight, ASD—autism spectrum disorder, pCNV—pathogenic copy number variations, LP—likely-pathogenic, VOUS—Variant of unknown significance, VOUS no classification—VOUS with no-subclassification.

**Table 1 healthcare-08-00305-t001:** Characteristics of 59 infants with developmental delay who included in this study.

Parameters	N	Min	Max	Mean	STD
Age (months)	59	8.3	84	38.7	18.50
0–12 months	3 (5.1%)	-	-	-	-
12–24 months	8 (13.6%)	-	-	-	-
24–36 months	17 (28.8%)	-	-	-	-
36–48 months	10 (17.0%)	-	-	-	-
48–60 months	10 (17.0%)	-	-	-	-
60–72 months	7 (11.9%)	-	-	-	-
72–84 months	4 (6.8%)	-	-	-	-
Birth weight	59	920	4300	2921.0	716.48
Gestational age	59	28	41	37.4	2.72
Gender					
Male	41 (69.5%)	-	-	-	-
Female	18 (30.5%)	-	-	-	-
Age of father at birth	59	24	52	35.7	5.23
Age of mother at birth	59	23	40	32.2	4.04
Delivery methods					
Vaginal delivery	23 (39%)	-	-	-	-
Caesarean section	36 (61%)	-	-	-	-
Mode of conception					
Natural pregnancy	50 (84.7%)				
Intra-uterine insemination	2 (3.4%)				
In vitro fertilization	7 (11.9)				
Comorbidity					
Cerebral palsy	1 (1.7%)	-	-	-	-
Autism Spectrum Disorder	10 (16.9%)	-	-	-	-
Cardiac anormaly	5 (8.5%)	-	-	-	-
Dysmorphism	2 (3.4%)	-	-	-	-
Brain anormaly	1 (1.7%)	-	-	-	-
Presence of any anormaly	5 (18.7%)	-	-	-	-

Min; minimum, Max; maximum, STD; standard deviation.

**Table 2 healthcare-08-00305-t002:** Clinical features and parameters with chromosomal microarray (CMA) in 35 infants with developmental delay and abnormalities in copy number variations.

No	Sex	Age (m)	Result of CMA	OMIM Data Base	OMIM Gene	Clinical Features	Age of Father	Age of Mother	Birth Weight	Delivery	GA
1	M	28.6	Yq11.221q11.222 duplication 1.5 Mb	Likely benign CNV	XKRY, CDY2A, HSFY1	DD	33	29	3350	C-sec	38
2	F	14.3	2q13 duplication 861 Kb	VOUS	RGPD6, MALL, NPHP1	DD	49	24	2600	C-sec	38
3	F	77.3	8p23.2 duplication 2.4 Mb	VOUS	CSMD1	DD, AuSD	34	33	3790	NVD	40
4	M	42	Yq11.223q11.23 duplication 3.1 Mb	Likely benign CNV	Multiple OMIM gene	DD, AuSD	47	36	3010	C-sec	39
5	M	15.1	10q11.22 duplication 1.3 Mb	Likely benign CNV	SYT15, GPRIN2, NPY4R	DD, Polydactyly	34	36	3900	C-sec	40
6	M	20.6	Yq11.223q11.23 duplication 1.6 Mb	Likely benign CNV	Multiple OMIM gene	DD, Umblicus fistula to intestine	37	34	2500	C-sec	39
7	M	70.7	10p15.3 deletion 1.4 Mb	VOUS	DIP2C, IDI2, IDI2-AS1, ADARB2	DD	38	33	2800	NVD	36
8	M	54.3	X121.2 deletion 523 Kb	Likely benign CNV	DACH2	DD	45	37	3150	NVD	40
9	F	24.6	2p16.3 deletion 142 Kb	Likely pathogenic CNV	NRXN1	DD, Occipital meningocele, RDS	41	36	2100	C-sec	34
10	M	34.2	7p21.1 duplication 484 Kb Xq21.33 duplication 611 Kb	Likely benign CNV VOUS	AHR DIAPH2, RPA4	DD	30	28	2500	C-sec	38
11	M	51.5	2q13 deletion 861 Kb Yq11.223q11.23 duplication 4.0 Mb	VOUS Likely benign CNV	RGPD6, MALL, NPHP1 Multiple OMIM gene	DD	33	32	3200	C-sec	38
12	M	46	1q43q44 duplication 5.8 Mb	VOUS	Multiple OMIM gene	DD, Hydronephrosis, Placenta previa	34	34	2920	NVD	38
13	M	35.6	3q29 duplication 1.7 Mb 15q13.3 duplication 440 Kb	Likely pathogenic CNV VOUS	Multiple OMIM gene CHRNA7	DD, AuSD	28	24	4180	NVD	37
14	F	42.9	Xp22.33 duplication 413 Kb	Likely benign CNV	GTPBP6, PPP2R3B	DD, AuSD	33	31	3990	C-sec	41
15	M	32.2	2q13 duplication 861 Kb	VOUS	RGPD6, MALL, NPHP1	DD, RDS	41	37	920	C-sec	30
16	M	14.1	7q11.21 deletion 443 Kb	Likely benign CNV	ZNF92	DD, PDA, RDS	32	29	1380	C-sec	28
17	M	14.1	10q11.22 duplication 1.1 Mb	Likely benign CNV	NPY4R	CP, VSD, RDS, Neonatal jaundice	33	33	1120	NVD	30
18	M	72.1	9q31.1 duplication 497 Kb Xp22.33 duplication 436 Kb	VOUS VOUS	SMC2 SHOX	DD, Neonatal jaundice	38	35	2500	NVD	38
19	M	41.3	10q11.22 deletion 1.1 Mb	Likely benign CNV	NPY4R	DD, AuSD	36	36	3660	C-sec	41
20	M	10.8	7q11.21 deletion 456 Kb Yq11.221q11.222 duplication 1.5 Mb	Likely benign CNV Likely benign CNV	ZNF92 XKRY, CDY2A, HSFY1	DD, RDS	29	28	2610	C-sec	37
21	M	51.1	2p24.3 duplication 814 Kb 11q25 duplication 477 Kb	Likely benign CNV Likely benign CNV	B3GAT1	DD, ID	36	31	3200	NVD	38
22	M	31.3	2q23.3 deletion 467 Kb Duplication, overall area of X chromosome	Benign CNV Pathogenic CNV(Klinefelter syndrome)	Multiple OMIM gene	DD	32	32	2500	C-sec	38
23	M	31	1p32.3 duplication 461 Kb	Likely benign CNV	ACOT11, TTC4, PARS2, DHCR24	DD	31	29	2700	NVD	38
24	F	22.9	3p26.3p26.1 duplication 2.2 Mb Loss mosaicism, overall area of X chromosome	VOUS Pathogenic CNV(Turner syndrome)	CNTN4, IL5RA, TRNT1 Multiple OMIM gene	DD, Neonatal jaundice	38	35	2700	C-sec	38
25	M	30.3	3q26.31 duplication 706 Kb 17p13.3 duplication 210 Kb	VOUS VOUS	NLGN1, NAALADL2 ABR, BHLHA9, TUSC5, YWHAE	DD, AuSD	36	27	2650	NVD	38
26	M	37.3	17p13.3 duplication 306 Kb	VOUS	ABR, BHLHA9, TUSC5	DD	34	33	2600	C-sec	38
27	M	29	2p25.3 duplication 133 Kb 2p25.3 duplication 191 Kb	VOUS VOUS	SNTG2 PXDN, MYT1L	DD	40	36	2660	C-sec	37
28	F	29	2p25.3 duplication 139 Kb 2p25.3 duplication 191 Kb	VOUS VOUS	SNTG2 PXDN, MYT1L	DD	40	36	3400	C-sec	37
29	F	10	2q37.1q37.3 deletion 8.8 Mb	Pathogenic CNV	Multiple OMIM gene	DD	39	36	3700	NVD	39
30	M	51.9	2q13 duplication 861 Kb 5p15.33 duplication 592 Kb Yq11.222 duplication 428 Kb	VOUS Likely benign CNV Likely benign CNV	RGPD6, MALL, NPHP1 IRX1 HSFY1	DD, AuSD	35	33	3260	C-sec	41
31	M	8.3	LOH, overall area of autosome 22q11.21 duplication 3.3 Mb Yq11.222 duplication 823 Kb	VOUS Pathogenic CNV VOUS	Multiple OMIM gene USP18, DGCR6, PRODH, DGCR2, DGCR14	DD, ASD, Megalencephaly, Soft claft palate	24	23	3200	NVD	38
32	F	49.6	10q11.22 duplication 1.2 Mb	Benign CNV	SYT15, GPRIN2, NPY4R	DD	38	31	3700	NVD	40
33	F	36.2	2q13 duplication 861 Kb	VOUS	RGPD6, MALL, NPHP1	DD	35	33	3700	NVD	38
34	F	71.1	10q11.22 duplication 1.1 Mb	Benign CNV	NPY4R	DD	33	32	3080	C-sec	38
35	M	26.8	2q11.2 duplication 431 Kb	Benign CNV	-	DD	39	39	2900	C-sec	36

CMA: chromosomal microarray, OMIM: online mendelian inheritance in man, GA: gestational age, VOUS: variants of uncertain significance, DD: developmental delay, C-sec: cesarean section, NVD: normal vaginal delivary, AuSD: autism spectrum disorder, RDS: respiratory distress syndrome, PDA: patent ductus arteriosus, LOH: loss of heterozygosity.

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
