# Peer review of "Relationship between Clinical Parameters and Chromosomal Microarray Data in Infants with Developmental Delay"

_healthcare, 2020, doi:10.3390/healthcare8030305_

Round 1

Reviewer 1 Report

In this article authors investigated if there is any significant correlation between CMA results from infants with developmental delays and various clinical parameters. Since authors have included most of the clinical parameters of the parents, it will be useful to also include the mode of conception, whether it was natural or using artificial reproduction techniques. I did not find any figures in the file I received. Thus, is it not possible to evaluate the article for scientific soundness. Tables were also not presented appropriately. 

Author Response

Reviewer 1

Comments and Suggestions for Authors

In this article authors investigated if there is any significant correlation between CMA results from infants with developmental delays and various clinical parameters. Since authors have included most of the clinical parameters of the parents, it will be useful to also include the mode of conception, whether it was natural or using artificial reproduction techniques. I did not find any figures in the file I received. Thus, is it not possible to evaluate the article for scientific soundness. Tables were also not presented appropriately.

Answer: We appreciate your valuable comment. We totally agree with your comment. Following your comment, we have added the fertilization methods (natural, intra-uterine insemination, and IVF). However, there was no statistical significant differences according to fertilization methods between pathogenic, VOUS-likely pathogenic, and VOUS with no sub-classification versus VOUS-likely benign, benign, and normal CNVs groups, and between pathogenic and VOUS-likely pathogenic CNV groups compared with VOUS with no sub-classification, VOUS-likely benign, benign, and normal CNVs groups, except for brain anomalies. We also modified figure 2 & 3, and tables. Thank you.

Reviewer 2 Report

  1. On page 3, the author states: “Recently, CMA has been widely used as a first-tier of genetic analysis in place of G-banded karyotyping as it can examine gene copy number variations (CNVs) throughout the genome with improved sensitivity for submicroscopic deletions and duplications. However, this diagnostic improvement is associated with a significant economic burden to the caregivers.” From this statement, CMA provides a great help. The author continues to mention: “Nevertheless, Farooqi et al. showed that the diagnostic yield of CMAs, with identification of pathogenic CNVs or variants of unknown significance (VOUS)-likely pathogenic, was less than 10% in infants with DD.[11] This finding highlights the clear need of a better understanding of DD clinical features and its association with CMA data to reduce unnecessary CMA testing.” If it also states the importance of CMS, the author needs more explanation.
  1. On page 4, do the 59 infants in this study refer to the ‘further evaluation’ or ‘follow-up test’ groups of K-DST results? Why is or instead of and? And how many infants are there in each of these two categories? It needs more clarification.
  2. The research method section did not explain how to implement this research, and the CMA protocol section did not explain as well.
  3. In the research results, the words on the figures are very vague and almost unrecognizable, and some of the data in the tables cannot be read.
  4. The presentation of descriptive statistical analysis should be more detailed. Since there are the numbers of male and female infants, the contents in Table 1 can also be separately stated. In addition, the age of children is actually quite different, and it is recommended that they should be presented in groups. In the K-DST results, how many people there are in each of the further evaluation group and the follow-up test group, the number of boys and girls, and the age should also be clearly explained.
  5. On page 6, the author mentioned: "we also evaluated potential correlations between demographic and clinical parameters of the infants and their parents, including risk factors known to cause DD, and the genetic findings." But it seems that the results of the correlation analysis did not appear.

Author Response

Reviewer 2

On page 3, the author states: “Recently, CMA has been widely used as a first-tier of genetic analysis in place of G-banded karyotyping as it can examine gene copy number variations (CNVs) throughout the genome with improved sensitivity for submicroscopic deletions and duplications. However, this diagnostic improvement is associated with a significant economic burden to the caregivers.” From this statement, CMA provides a great help. The author continues to mention: “Nevertheless, Farooqi et al. showed that the diagnostic yield of CMAs, with identification of pathogenic CNVs or variants of unknown significance (VOUS)-likely pathogenic, was less than 10% in infants with DD.[11] This finding highlights the clear need of a better understanding of DD clinical features and its association with CMA data to reduce unnecessary CMA testing.” If it also states the importance of CMS, the author needs more explanation.

Answer: We appreciate your valuable comment. We totally agree with your comment. In those sentences, we`d like to emphasize the importance of reducing unnecessary CMA testing, not the importance of CMA.

On page 4, do the 59 infants in this study refer to the ‘further evaluation’ or ‘follow-up test’ groups of K-DST results? Why is or instead of and? And how many infants are there in each of these two categories? It needs more clarification.

Answer: We appreciate your valuable comment. We totally agree with your comment. Following your comment, we have added more explanation in the manuscript as follows;

“All 59 infants with DD who included in this study were classified within the ‘further evaluation’ or ‘follow-up test’ groups according to K-DST (‘further evaluation’; 23 infants, ‘follow-up test’; 36 infants).”

The research method section did not explain how to implement this research, and the CMA protocol section did not explain as well.

In the research results, the words on the figures are very vague and almost unrecognizable, and some of the data in the tables cannot be read.

Answer: We appreciate your valuable comment. We totally agree with your comment. Following your comment, we have modified figures and tables. In PDF version, there may be some errors contrast to word version.

The presentation of descriptive statistical analysis should be more detailed. Since there are the numbers of male and female infants, the contents in Table 1 can also be separately stated. In addition, the age of children is actually quite different, and it is recommended that they should be presented in groups. In the K-DST results, how many people there are in each of the further evaluation group and the follow-up test group, the number of boys and girls, and the age should also be clearly explained.

Answer: We appreciate your valuable comment. We totally agree with your comment. Following your comment, we have modified table 1. We also added an explanation in the manuscript as follows;

“All 59 infants with DD who included in this study were classified within the ‘further evaluation’ or ‘follow-up test’ groups according to K-DST (‘further evaluation’; 23 infants, ‘follow-up test’; 36 infants).”

On page 6, the author mentioned: "we also evaluated potential correlations between demographic and clinical parameters of the infants and their parents, including risk factors known to cause DD, and the genetic findings." But it seems that the results of the correlation analysis did not appear.

Answer: We appreciate your valuable comment. We totally agree with your comment. Following your comment, we have added it in figure 2 & 3.

Reviewer 3 Report

An interesting and overall well written paper which I believe has some important messages to convey.

I have added my comments to the uploaded file.

Author Response

We really appreciate your valuable comment. 

We added our modification in PDF file.

Thank you.

Reviewer 4 Report

The study by Lee et. al aim  to investigate possible clinical relashionship between developmental delay and chromosomal microarray data. The paer irs well written and mainly report negative data.

Line 25: VOUS:full meaning of the word should be included

Line 155- 156: This is an important point. The technical limitations of CMA should be a bit more explained. Particularly in case of DD and comorbidity.

Sharpness of Fig. 2and 3 should be  highly improved.

Author Response

Reviewer 4

The study by Lee et. al aim  to investigate possible clinical relashionship between developmental delay and chromosomal microarray data. The paer irs well written and mainly report negative data.

Line 25: VOUS:full meaning of the word should be included

Answer: We appreciate your valuable comment. We totally agree with your comment. Following your comment, we have modified it as follows;

Except for brain anomalies, there was no statistically significant differences between infants who had pathogenic and variants of unknown significance (VOUS)-likely pathogenic CNVs groups compared with those within the VOUS-likely no sub-classification, VOUS-likely benign, benign, and normal CNVs groups.”

Line 155- 156: This is an important point. The technical limitations of CMA should be a bit more explained. Particularly in case of DD and comorbidity.

Answer: We appreciate your valuable comment. We totally agree with your comment. Following your comment, we have added an more explanation in the manuscript as follows;

For example, CMA cannot detect balanced chromosome rearrangements, such as inversions or translocations, which do not result in deletion or duplication of genetic material, or cases of low-level tissue mosaicism, although balanced rearrangements rarely are associated with disease unless there is disruption of a critical gene. Additionally, CMA may not identify low levels of tissue mosaicism in the fetus.”

Sharpness of Fig. 2and 3 should be highly improved.

Answer: We appreciate your valuable comment. We totally agree with your comment. Following your comment, we have modified it

Round 2

Reviewer 1 Report

Authors appropriately responded to previous comments and provided the tables and figures. This study have multiple limitations including small sample size and technical limitations of CMA. However, authors have discussed these limitations.

Author Response

Authors appropriately responded to previous comments and provided the tables and figures. This study have multiple limitations including small sample size and technical limitations of CMA. However, authors have discussed these limitations.

Answer: We appreciate your valuable review comment. As your comment, we have added it in the limitation part of our manuscript. Thank you.

Reviewer 2 Report

The author has completed the revision based on the review comments.

Author Response

The author has completed the revision based on the review comments.

Answer: We really appreciate your valuable comment. Thank you.